# LAST: Latent Structure guided Gaussian Splatting from Monocular Human Videos

## Abstract

Multiocular human reconstruction aims to create a high-quality 3D human representation from sparse video data. Recently, 3D Gaussian Splatting (3DGS) has demonstrated impressive results in multiocular human reconstruction tasks, exhibiting remarkable speed and accuracy. However, it encounters challenges in scenarios involving intricate clothing and dynamic postures. This problem may stem from pixel-level supervision during the 3DGS optimization process, which probably lead to spurious associations between unrelated visual features (*e.g.,* misinterpreting clothing wrinkles as dependent on body occlusions rather than recognizing that both are influenced by complex postures). To address this issue, we propose the LAST framework for realistic 3D human reconstruction, which integrates a pre-trained Image-to-Point (I2P) model to enhance the 3D Gaussian Splatting optimization pipeline. The core of the LAST is to disentangle meaningful latent factors and realistic dependencies from the input video frames, which allows for dynamic adjustments to the density and attributes of Gaussian points during the optimization process. Experimental results demonstrate that our method significantly improves accuracy and realism in 3D human reconstruction compared to existing techniques, particularly in challenging scenarios involving complex posture and intricate clothing details.

## 1 Introduction

3D human reconstruction is a critical task in computer vision with broad applications in virtual reality and game development. Traditional 3D human reconstruction methods (Bradley et al., 2008; Collet et al., 2015; Starck & Hilton, 2007; Guo et al., 2019) often rely on images from dense viewpoints captured by numerous cameras. However, such methods face substantial limitations in terms of hardware costs and the complexity of data collection. Recently, advanced methods (Weng et al., 2022; Jiang et al., 2022b; Yu et al., 2023; Peng et al., 2021a) have demonstrated that high-quality human reconstruction can be achieved from sparse-view images using Neural Radiance Fields (NeRF) representations. However, these methods typically require expensive time and computational costs in the training and rendering, making them difficult to apply in the real world.

Recent advances in 3D Gaussian Splatting (3DGS, Kerbl et al. (2023)), it is possible to achieve high expressivity with significantly faster training and rendering speeds compared to NeRF-based methods. While recent works of 3DGS human reconstruction (Moreau et al., 2024b; Qian et al., 2024; Hu et al., 2024b; Li et al., 2024a) perform well in most scenarios, they struggle with challenging poses and delicate clothing, suffering from detail loss and distortion. This is because 3DGS encodes latent factors that influence 3D human reconstruction, such as lighting, clothing, and pose, into Gaussian points, which have dependency relationships between each other. For instance, variations in posture can affect clothing details including folds or stretches, it's essential to account for these dependencies during the optimization process. However current methods directly use the difference between rendered and real images to guide model optimization. Images contain highly coupled visual information (Liu et al., 2022b), with their dependencies being messy and intertwined. When learning dependencies based on statistical relationships, probably leads to misleading conclusions, which are analogous to the spurious correlations in the causal theory (Eberhardt & Scheines, 2007). For example, wrinkled clothing frequently occurs at the same time as body occlusion in the athletic position, the optimization process might incorrectly assume that body occlusion depends on wrinkled clothing, but such dependencies are not routinely valid in reality, as both scenarios depend on complex

posture. If spurious correlations are incorporated into the optimization process, latent factors will interfere with each other during the optimization process. This interference results in a detrimental effect where adjustments in one latent factor can inadvertently cause degradation in others, manifesting as artifacts or loss of fine details in the final reconstruction. This issue is particularly pronounced in challenging poses and delicate clothing, which exhibit more complex dependency relationships.

To solve the above problems, we argue that two main goals should be achieved: (1) discover latent factors and their corresponding dependencies within the 3D model; and (2) control the 3DGS optimization process to mitigate the mutual interference among different latent factors. For the first goal, we propose a pre-training Image-to-Point (I2P) framework, which utilizes a Variational Autoencoder (VAE, Kingma (2013)) to learn how to disentangle 2D images into semantic vectors representing latent factors, and we design a **latent structure** that focuses on 3D reconstruction and incorporates it into the I2P framework. This latent structure aims to learn a dependency matrix that represents dependencies between latent factors. During the VAE training process, features of latent factors are extracted from the encoder and propagated from the parent latent factor to the child latent factor, the decoder then uses these propagated features to generate the point cloud, enabling a backpropagation process that simultaneously updates the parameters of the VAE and the dependency matrix. This training process generates latent representations with explicit semantics and dependency associations, which serve as additional supervisory signals and guidance of the 3DGS optimization process to mitigate latent factor mutual interference.

For the second goal, traditional 3DGS optimization (Li et al., 2024a; Shao et al., 2024; Abdal et al., 2024b) relies on image difference loss gradients to guide the densification and updating of Gaussian point attributes. However, this can inadvertently propagate potentially incorrect dependencies due to the image coupling mentioned above. To address this, we propose to incorporate the learned latent factors and dependency matrix from the I2P framework. To be specific, there are three core strategies: (1) decouple latent factors from images as new supervisory signals in the optimization process. Compared to the pixel supervision, this allows for a clearer understanding of how each latent factor influences the reconstruction; (2) utilize the decoder to directly transform the latent factors space into the point cloud space, establishing a connection between the 3DGS and the latent factors. By analyzing differences in latent factors, we can identify 3DGS critical regions that require updates. (3) construct an updated sequential chain based on the topological order indicated by the dependency matrix to minimize the risk of interference among latent factors. Combining these strategies not only leverages the latent factors but also reduces the propagation of erroneous dependencies. We evaluate our approach on standard benchmarks for 3D human reconstruction, comparing it against state-of-the-art methods. Our experiments show that our model achieves superior reconstruction quality, particularly in challenging scenarios involving complex poses and intricate clothing details.

Overall, this work makes the following contributions:

1. We introduce an approach to disentangle and learn the dependencies among latent factors from 2D visual information, which allows us to manage the complex interplay between factors such as posture, clothing, and lighting.

2. We introduce the dependencies between latent factors into the 3DGS optimization process to mitigate mutual interference among latent factors. Empirically, this integration allows for a more nuanced optimization strategy that maintains the fidelity of local detail while ensuring overall coherence in the reconstructed model.

3. We evaluate the proposed method on the ZJU-MoCap, MoCap, and DNA-Rendering datasets. Across all datasets, our approach achieves state-of-the-art performance in rendering quality. Furthermore, detailed ablations are conducted to demonstrate the effectiveness of the proposed components.

## 2 RELATED WORK

### 2.1 3D DIGITAL HUMAN RECONSTRUCTION

Previous studies (Alldieck et al., 2022; Choi et al., 2022; Halimi et al., 2022; Habermann et al., 2021) often use point clouds or meshes as the output 3D representations. While they incorporate latent factors such as light fields and dynamic textures, fitting these priors to fine deformations and

texture details remains challenging. Neural Radiance Fields (NeRF) (Zheng et al., 2023; Hong et al., 2022; Xu et al., 2021; Jiang et al., 2022a; Hu et al., 2023)models the radiance (color and density) of each point in the scene as a neural network, turning the learning of latent factor priors into a dense regression task, which addresses some of these issues. However, NeRF models typically require pre-training on clear human data and fine-tuning for new human performers, which is inefficient and may take several hours of pre-training to obtain a 3D human representation.

3D Gaussian Splatting (Hu et al., 2024a; Moreau et al., 2024a; Li et al., 2024b; Zheng et al., 2024; Abdal et al., 2024a) offers a more efficient method to generate high-quality 3D human representations by mapping pixels from 2D images to Gaussian points in 3D space. It decouples latent factors of 3D space into attributes of Gaussian points in the form of Gaussian distributions, enabling efficient and accurate human reconstruction. In this work, we focus on dependency relationships among latent factors that previous research has not considered.

## 2.2 CAUSALITY IN VISION

Current research has made strides in integrating visual features with causal inference to learn visual representations (Wang et al., 2020; Lopez-Paz et al., 2017; Chalupka et al., 2014; Liu et al., 2022c;a; Wang et al., 2021; Zareian et al., 2020), enhancing models' understanding of objects and their relationships within images, and improving performance on downstream tasks such as object detection, image classification, and visual question answering. Another aspect of research involves introducing causal relationships to address challenges in traditional visual tasks. For instance, Zhang et al. (2020) improved semantic segmentation quality by severing the causal link between contextual priors and images.Qi et al. (2020) enhanced the accuracy of dialogue systems by using questions as intermediaries and cutting direct causal effects between dialogue history and answers. Yang et al. (2021) improved image annotation accuracy by clarifying the causal relationships among image features, potential confounding factors, and image labels. However, the existing methods lack research in 3D visual information. Inspired by causal structure learning, we proposed latent structure learning to discover dependencies of latent factors and integrate them into the 3D Gaussian Splatting optimization process for 3D human reconstruction.

## 3 METHOD

LAST aims to achieve realistic 3D human representation from sparse video data. As shown in Figure. 1, the overall pipeline is composed of a pre-trained image-to-plane (I2P) model and a enhanced 3D Gaussian Splatting optimization pipeline, which is informed by the prior knowledge acquired from the I2P model.

### 3.1 IMAGE-TO-POINT MODEL FOR IMAGE DISENTANGLEMENT

In the first stage, we need to reconstruct a coarse 3D human representation $\mathbf{R}$ of the from a 2D image $X$. Specifically, we learn a feed-forward network that directly transforms the input image into the point cloud representation, namely the Image-to-Point (I2P) model. When designing the network structure of the I2P model, there are two important tasks for the network: (1) It should disentangle latent factors representing 3D semantic information from observed 2D images. (2) It should learn the correct dependencies from the chaotic dependencies of visual features. To this end, as shown in Figure 1, we design a VAE-based hybrid network consisting of **Latent Factor Disentanglement** module and **Latent Structure Learning** module.

### 3.1.1 LATENT FACTOR DISENTANGLEMENT

Latent factor disentanglement aims to decompose the observational image into semantically distinct dense vectors, where each vector represents a key aspect influencing the outcome of 3D human reconstruction. We follow the Variational Autoencoder (VAE) paradigm to disentangle the latent factors contained in 2D images. In our framework, we generate a point cloud $\mathbf{R}$ from latent factors $\mathbb{Z}$ disentangled from the observed data $X$. Formally, considering the generative model $p_\theta(\mathbf{R}|\mathbb{Z})$ and the prior distribution $p_\theta(\mathbb{Z})$, our goal is to infer the posterior distribution $p_\theta(\mathbb{Z}|\mathbf{R})$ to utilize latent space for reconstruction of 3D human point cloud. However, both the integral of marginal likelihood

$p_\theta(\mathbf{R}|\mathbb{Z})$ and the posterior density $p_\theta(\mathbb{Z}|\mathbf{R})$ are intractable. Therefore, We approximate a variational posterior $q_\phi(\mathbb{Z}|\boldsymbol{X})$ from images with similar distributions to the unknown true posterior $p_\theta(\mathbb{Z}|\mathbf{R})$. The Evidence Lower Bound (ELBO) is derived to optimize the model:

$$\mathcal{L}(\theta, \phi; \boldsymbol{X}) = \mathbb{E}_{q_\phi}[\log p_\theta(\mathbf{R}|\mathbb{Z})] - D_{\mathrm{KL}}(q_\phi(\mathbb{Z}|\boldsymbol{X})\|p_\theta(\mathbb{Z})), \tag{1}$$

where $\theta$ is the parameter of the generation model which maps latent spaces from 2D images, $\phi$ is the parameter of the inference model which reconstructs 3D human representation from latent spaces, $D_{\mathrm{KL}}(q_\phi(\mathbb{Z}|\boldsymbol{X})\|p(\mathbb{Z}))$ is the Kullback-Leibler divergence between the approximate posterior and the conditional prior.

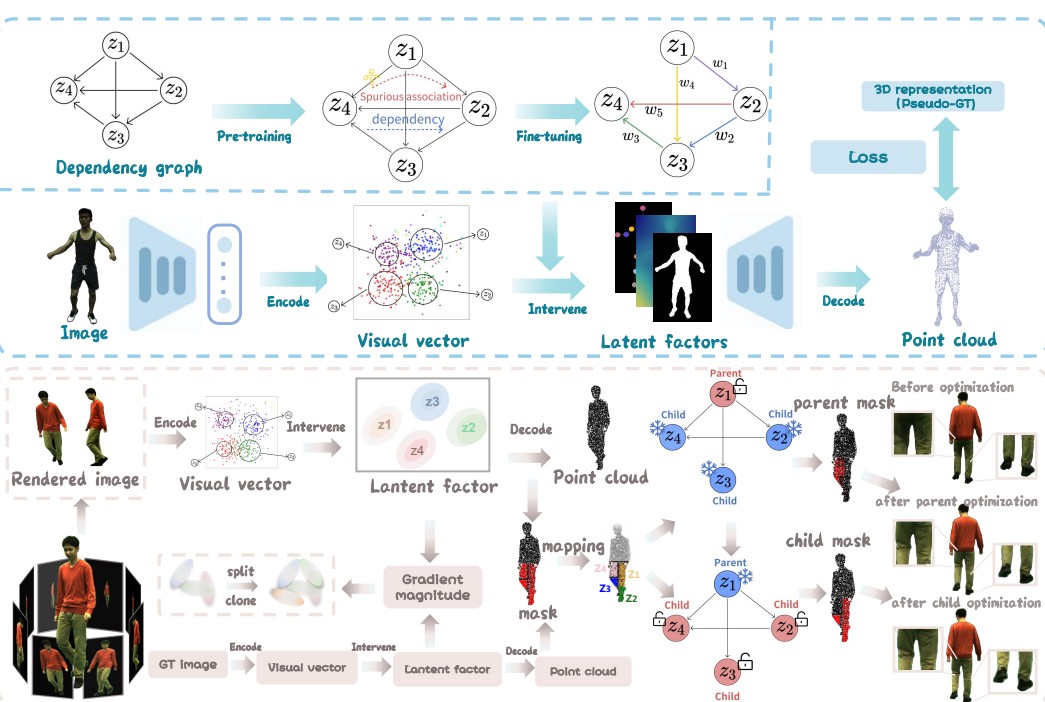

Figure 1: The network structure of I2P model and improved 3DGS optimization pipeline.

In the implementation of our framework, we design the architecture consisting of an encoder and a decoder. The encoder maps the input image $\boldsymbol{X}$ into the latent space $\mathbb{Z}$, producing parameters for a Gaussian distribution—namely, the mean and variance—thereby allowing the formulation of the variational posterior $q_\phi(\mathbb{Z}|\boldsymbol{X})$. We employ the reparameterization trick to sample from this Gaussian distribution, enabling the backpropagation of gradients. The decoder subsequently reconstructs the 3D representation $\mathbf{R}$ from $\mathbb{Z}$, outputting a probability distribution over the reconstructed result. During training, we train the encoder and decoder by minimizing the ELBO loss function and performing backpropagation to update the parameters $\theta$ and $\phi$.

### 3.1.2 LATENT STRUCTURE LEARNING

Once the latent factors are disentangled from 2D images, the second question is how to learn real dependencies from chaotic dependencies of visual features that contain spurious correlations. Formally, spurious correlations can be denoted as a latent factor $z_k$ that affects both the other latent factor $z_i$ and $z_j$. There will be a conditional probability relationship between $z_i$ and $z_j$, but there is no relevance between them in real semantics, resulting in a distorted relationship between $z_i$ and $z_j$. Inspired by Structural Causal Model(SCM,Shimizu et al. (2006)), we incorporate intervention into the VAE training process to solve the above problem. The intervention operation aims to fix the value of the latent factor $z_k$ to cut off its influence on other factors, allowing us to handle relationship the between the $z_i$ and $z_j$ in isolation.

In the implementation, we incorporate Latent Structure Learning between the existing encoder and decoder of the VAE to simulate the intervention of latent factors. Latent Structure Learning is a

learnable matrix to represent dependencies among latent factors. In the training process, we set a arbitrary latent factor to a fixed value and propagate this change to all subordinate latent factors according to the dependency matrix. This allows us to generate expected counterfactual outputs and compare them with the ground truth to update the parameters of the learnable matrix for capturing the correct dependencies. Since the latent factors disentangled from images using the encoder often do not account for the realistic dependency relationships, therefore we take exogenous latent factors sampled from visual features as input of Latent Structure Learning, and output endogenous latent factors containing realistic dependencies. Formally, this process can be represented as follows:

$$\mathbf{Z} = \boldsymbol{A}^T \mathbf{Z} + \epsilon = \left(\boldsymbol{I} - \boldsymbol{A}^T\right)^{-1} \epsilon, \epsilon \sim q_\phi(\mathbb{Z}|\boldsymbol{X}), \tag{2}$$

where $\boldsymbol{A}$ denotes the trainable matrix where each element represents the dependency between a particular pair of latent factors. $\epsilon$ denotes exogenous latent factors sample from $q_\phi(\mathbb{Z}|\boldsymbol{X})$ and $\mathbf{Z}$ denotes endogenous latent factors as output which contains the semantics and dependencies of 3D representation. In practice, we introduce a neural network $g_\eta(\cdot)$ to propagate the dependency of different latent factors by mask mechanism (Ng et al., 2022).

$$\hat{\boldsymbol{z}}_i = g_\eta(\boldsymbol{A}_i \circ \mathbf{Z}) + \epsilon_i, \tag{3}$$

where $\circ$ is the element-wise multiplication, $\boldsymbol{A}_i \circ \mathbf{Z}$ equals to a vector that only contains its parental information as it masks all non-parent latent factors from $\mathbf{z}_i$. The final 3D human representation is reconstructed based on $\{\hat{\boldsymbol{z}}_1, \hat{\boldsymbol{z}}_2, ..., \hat{\boldsymbol{z}}_n\}$, which encodes latent factors and their dependencies with non-linear transformations. Parameters $\boldsymbol{A}$ and $\eta$ will be updated in backpropagation. To improve dependency identification, we resort to causal discovery and propose three regularization objectives for training $\boldsymbol{A}$ as follows:

*Reconstruction Regularization:* The goal of reconstruction regularization is to ensure that the learned latent factors and their dependencies can accurately reconstruct the 3D human. Through this regularization, we provide not only a stable objective function for the VAE parameter optimization but also a proxy task for the dependency matrix optimization, indirectly assessing whether correct dependencies are learned, thereby achieving the overall goals of bi-level optimization:

$$\mathcal{L}_{\text{rec}} = \|\mathbf{R}_{gt} - \mathcal{D}(\mathbf{Z}, \boldsymbol{A})\|^2, \tag{4}$$

where $\mathbf{R}_{gt}$ represents the ground truth of 3D human representation, $\mathbf{Z}$ is the endogenous factors obtained from the input image, $\mathcal{D}(\cdot)$ represents the decoder that reconstructs the 3D representation based on the endogenous latent factors and dependency matrix $\boldsymbol{A}$.

*Disentanglement regularization:* The goal of disentanglement regularization is to ensure that each latent factor represents independent semantic information. In the VAE parameter optimization process, disentanglement regularization helps reduce redundancy and overlap among the latent factors of 2D images (exogenous factors) extracted by the encoder. In the dependency matrix optimization process, disentanglement regularization ensures that the transformation from exogenous factors to endogenous factors maintains the conditional independence:

$$\mathcal{L}_{\text{MI}} = \sum_{i<j} \int_{\hat{\boldsymbol{z}}_i} \int_{\hat{\boldsymbol{z}}_j} p(\hat{\boldsymbol{z}}_i, \hat{\boldsymbol{z}}_j) \log \frac{p(\hat{\boldsymbol{z}}_i, \hat{\boldsymbol{z}}_j)}{p(\hat{\boldsymbol{z}}_i)p(\hat{\boldsymbol{z}}_j)} \, d\hat{\boldsymbol{z}}_i \, d\hat{\boldsymbol{z}}_j. \tag{5}$$

*Dependency Regularization:* Dependency regularization is to ensure that the dependencies among latent factors align with the expected dependency structure. Specifically, dependency regularization enforces that any changes in parent latent factors influence only their child latent factors and do not affect other unrelated latent factors. This requirement implies that the relationships among latent factors should be represented as a Directed Acyclic Graph (DAG) to avoid cyclic dependencies:

$$\mathcal{L}_{dag} = \text{tr}\left((\boldsymbol{I} + \alpha \boldsymbol{A} * \boldsymbol{A})^n\right) - n, \tag{6}$$

where tr denotes the matrix trace and $\alpha$ is a hyper-parameter that depends on a prior estimation of the largest eigenvalue of $\boldsymbol{A} * \boldsymbol{A}$.

### 3.1.3 TRAINING

We propose to use a pre-trained and fine-tuning paradigm to train the I2P model. In the pre-training phase, we train the I2P model on a large human reconstruction dataset to capture dependencies that

are generally present across most human samples. There are three goals in this phase: (1) train the encoder to disentangle latent factors from 2d image. (2) construct correct dependency matrix $\boldsymbol{A}$ to mitigate spurious correlations of latent factors. (3) train the decoder to generate point clouds from representations of latent factors. In the fine-tuning phase, we adapt the pre-trained I2P model to specific downstream tasks. In this stage, we freeze the parameters of the encoder and decoder, fine-tuning weights in the dependency matrix $\boldsymbol{A}$ based on the input samples. This means that the structure of $\boldsymbol{A}$ will dynamically adjust according to the task, enhancing the model's ability to capture unique features in specific human samples. We adopt regularization objectives mentioned in sec 3.1.2 to train the I2P model in both the pre-trained phase and fine-tuning phase. Which is defined as follows:

$$\mathcal{L} = w_1 \mathcal{L}_{rec} + w_2 \mathcal{L}_{MI} + w_3 \mathcal{L}_{dag}. \tag{7}$$

where $\mathcal{L}_{rec}$ denotes the reconstruction regularization loss, $\mathcal{L}_{MI}$ denotes the disentanglement regularization, $\mathcal{L}_{dag}$ denotes the dependency regularization loss. The weights $w_1$, $w_2$, and $w_3$ are hyperparameters that balance the influence of these loss components.

## 3.2 LATENT STRUCTURE GUIDED GAUSSIAN OPTIMIZATION

With the pre-training process of the I2P model, we achieve to reconstruct an accurate 3D representation and dependency graph from the source image. In the following sections, we will discuss how to use the prior knowledge of the I2P model to enhance the performance of 3DGS optimization.

### 3.2.1 GAUSSIAN DENSITY ADAPTER

3DGS utilizes sparse point clouds generated by Structure from Motion (SfM) as initial input and subsequently employs an adaptive density control mechanism to refine the point cloud density, which is determined by the average magnitude of the gradient of the NDC coordinates for the viewpoints:

$$\frac{\sum_{m \in \mathbb{M}} \sqrt{\left(\frac{\partial \mathcal{L}_{2D,m}}{\partial \mu_{x,m}}\right)^2 + \left(\frac{\partial \mathcal{L}_{2D,m}}{\partial \mu_{y,m}}\right)^2}}{|\mathbb{M}|} > \tau_{pos}, \tag{8}$$

where $\mathbb{M}$ is the set of all viewpoints. Under the viewpoint $m \in \mathbb{M}$, the NDC coordinate is $(\mu^{x,m}, \mu^{y,m}, \mu^{z,m})$, and the loss between rendered image and ground truth under is $\mathcal{L}_{2D,m}$, and $\tau_{pos}$ is the threshold to determine whether a point is split or cloned.

However, this method relies on pixel-level supervision signals, which suffer from issues of spurious correlations. To address these challenges, we utilize the latent factors and their dependencies to guide the adaptive splitting or cloning of Gaussian points:

$$\frac{\sum_{m \in \mathbb{M}} \sqrt{\left(\frac{\partial \mathcal{L}_{rec,m}}{\partial \mu_{x,m}}\right)^2 + \left(\frac{\partial \mathcal{L}_{rec,m}}{\partial \mu_{y,m}}\right)^2 + \left(\frac{\partial \mathcal{L}_{rec,m}}{\partial \mu_{z,m}}\right)^2}}{|\mathbb{M}|} > \tau_{rec}, \tag{9}$$

where $\mathcal{L}_{rec,m}$ represents the difference between latent factors and their dependencies extracted from the rendered image and ground truth separately. The $\tau_{rec}$ is the new threshold to determine whether a point is split or cloned. This method transforms the supervisory signals for density control from the pixels of 2D images to the latent factors of the 3D model. Meanwhile, this integration allows us to adjust the point cloud density based on their dependencies rather than in isolation.

### 3.2.2 PROGRESSIVE UPDATE STRATEGY

The optimization in the 3DGS is based on successive iterations of comparing the rendered image to the training views in the captured dataset. This iterative process involves updating the attributes of 3D Gaussian points to minimize the discrepancy between the rendered image and the ground truth images from the dataset. The optimization process faces similar challenges as density control, primarily due to the coupling of pixel-level information and the risk of spurious correlations. Therefore, we propose a progressive update strategy to address the above problems. This methods explicitly introduce the dependence between latent factors into the optimization order of Gaussian points. Specifically, the strategy is as follows: (1) We use the I2P model to generate a point cloud $\boldsymbol{R}$ from the input image $\boldsymbol{X}$, which preserves a mapping relationship with 3DGS. We compare the

point clouds generated by the rendered images and the corresponding ground truth to identify key regions that require updates. We can translate these identified regions into key Gaussian points of each latent factor. (2) We establish an optimization sequence for the various latent factors based on the topology order in the dependency matrix $\boldsymbol{A}$. This sequence determines the order in which key Gaussian points are optimized. In each optimization iteration, start by updating the key Gaussian points of parent latent factors $\boldsymbol{z}_p$, the objective is to minimize the difference between the rendered image $\boldsymbol{X}_{rend}$ and the ground truth $\boldsymbol{X}_{gt}$ as global optimization:

$$\boldsymbol{z}_p^{(t+1)} = \arg\min_{\boldsymbol{z}_p} \mathcal{L}(\boldsymbol{X}_{rend}, \boldsymbol{X}_{gt}), \boldsymbol{X}_{rend} \in \mathcal{D}(\{\boldsymbol{z}_p, \boldsymbol{z}_s\}, \boldsymbol{A}), \quad (10)$$

where $\boldsymbol{z}_p$ denotes the parent latent factors, $\boldsymbol{z}_s$ represents the latent child factors, $t$ denotes number of iterations. After updating the key Gaussian points of parent latent factors $\boldsymbol{z}_p$, adjust all the key Gaussian points of child latent factors $\boldsymbol{z}_s$ that are dependent on $\boldsymbol{z}_p$. The objective is to minimize the rendering error while balancing the relationships between factors as local optimization:

$$\boldsymbol{z}_s^{(t+1)} = \arg\min_{\boldsymbol{z}_s} \mathcal{L}(\boldsymbol{X}_{rend}, \boldsymbol{X}_{gt}), \boldsymbol{X}_{rend} \in \mathcal{D}(\{\boldsymbol{z}_p^{t+1}, \boldsymbol{z}_s\}, \boldsymbol{A}). \quad (11)$$

Considering that local optimization does not excessively disturb global optimization, we set a threshold $\lambda$ for local optimization. This threshold serves as a safeguard to ensure that maintain the integrity of the overall model while allowing for necessary refinements in specific areas:

$$\boldsymbol{z}_s^{(t)} - \boldsymbol{z}_s^{(t+1)} = \nabla_{\boldsymbol{z}_s} \mathcal{L}(\boldsymbol{X}_{rend}, \boldsymbol{X}_{gt}) < \lambda, \boldsymbol{X}_{rend} \in \mathcal{D}(\{\boldsymbol{z}_p^{t+1}, \boldsymbol{z}_s\}, \boldsymbol{A}). \quad (12)$$

## 4 EXPERIMENT

### 4.1 EXPERIMENTAL SETUP

**Datasets.** We conduct experiments on three datasets: MonoCap (Peng et al., 2022),ZJU-MoCap (Jiang et al., 2023b), and DNA-Rendering (Cheng et al., 2023). MonoCap and ZJU-MoCap are conventional data sets for 3D reconstruction, which are used to compare our method with advanced methods. DNA-Rendering provides challenging scenes with complex poses and costumes.

**Comparison Methods.** We compare with six baselines. NeuralBody (Peng et al., 2021b) makes the first attempt to transform 3D human representation into latent code space. InstantAvatar (Jiang et al., 2023a), AnimatableNeRF (Peng et al., 2021a) are state-of-the-art methods for nerf-based human reconstruction. HuGS (Moreau et al., 2024b), Gauhuman (Hu et al., 2024b) and 3DGS-Avatar (Qian et al., 2024) are state-of-the-art methods for 3DGS-based humuan reconstruction.

### 4.2 QUALITATIVE EVALUATIONS

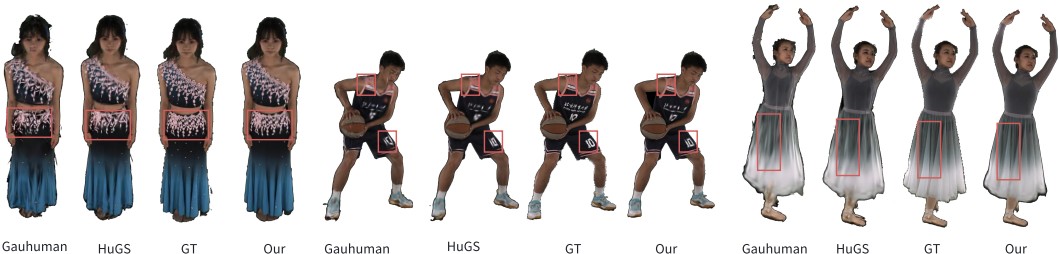

Gauhuman    HuGS    GT    Our     Gauhuman    HuGS    GT    Our     Gauhuman    HuGS    GT    Our

Figure 2: Comparison with 3DGS-based methods on the DNA-Rendering dataset. Despite complex motion and textured garments, our method preserves more details than other methods and can fit unusual joint deformation and clothes wrinkle.

**Evaluation on DNA-Rendering dataset:** As shown in Figure. 2, LAST is capable of generating high-quality renderings in complex clothes(*e.g.*, woman with national dress in the first case) and motion(*e.g.*, basketball player in the second case). Due to the inherent dependency association in our model design, LAST can allow for the preservation of fine textures and natural movements,

which are often challenging for traditional methods. Moreover, as shown in the third case, due to the consideration of the dependencies of different latent factors, our reconstruction results appear more natural at the fold of clothing. This design allows the posture to interact with the dynamic changes in the clothing, resulting in more delicate and realistic details.

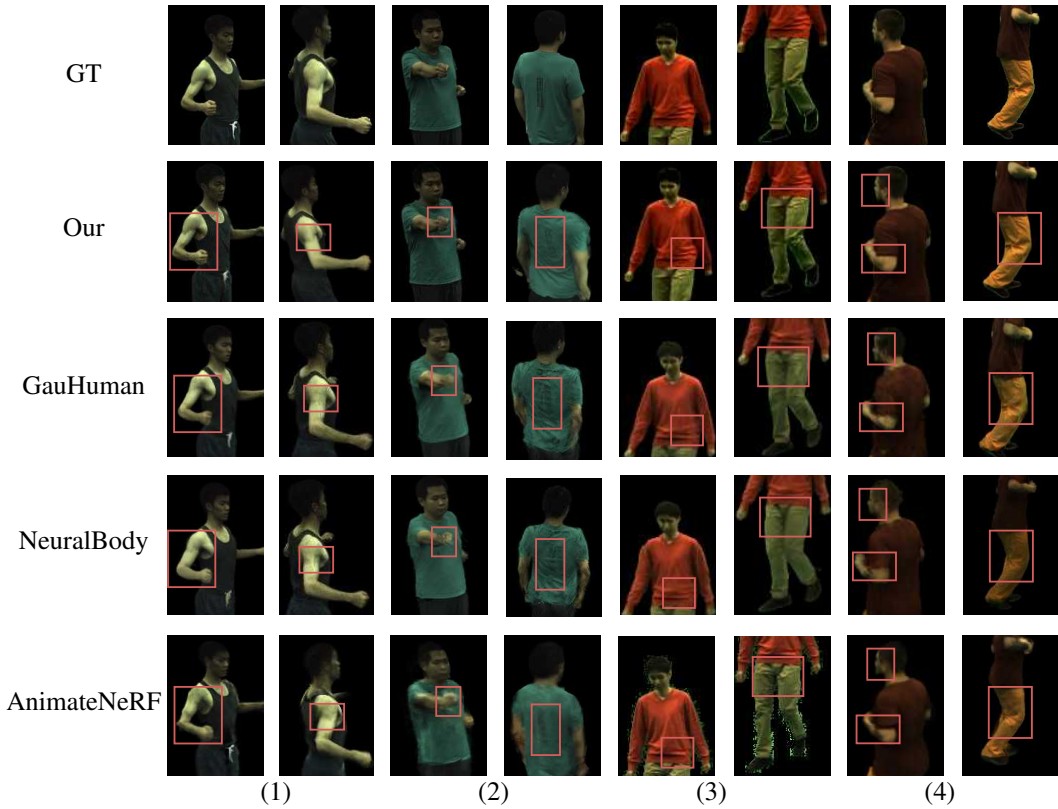

Figure 3: Qualitative comparison on the ZJU-MoCap and MonoCap datasets, showcasing the superior appearance and geometric details achieved by our method compared to baseline approaches.

**Evaluation on MonoCap and ZJU-MoCap dataset:** As shown in Figure. 3, our LAST exhibits superior appearance and geometric details than GauHuman, NeuralBody, and AnimateNeRF. Although GauHuman demonstrates a commendable ability to capture the human shape and posture, it often compromises on intricate details, leading to visible artifacts in certain regions (*e.g.,* obvious artifacts in the fist of case 2 and face of case 4). This discrepancy in detail fidelity might be attributed to the presence of spurious correlations that affect the optimization process. In GauHuman, the optimization is influenced by pixel-based supervision, which may struggle to balance the preservation of fine details with the overall results. Our method leverages the learned latent structure and dependency matrix to ensure that adjustments made during optimization are informed by the dependencies. This approach allows our model to preserve intricate details while managing complex interactions, resulting in smoother transitions and fewer artifacts. AnimateNeRF shows impressive results in case 3 and case 4 which contain intricate clothing textures and folds, but it exhibits limitations in reconstructing human posture (*e.g.,* significant arm distortion in case 1). This limitation might be attributed to the inefficient representation of humans in AnimateNeRF, which probably leads to high computational resource demands and incompletely captures the complexities of human poses. In contrast, our model disentangles latent factors of human reconstruction in challenging poses to employ a latent structure for human representation. NeuralBody optimizes the model directly in the latent space, but this updating method exhibits instability (*e.g.,* variations in arm reconstruction quality across different viewpoints in case 1). Our method employs progressive updates in 3DGS and ensures that modifications to the Gaussian points are based on learned latent structure, which helps maintain coherence across the generated outputs.

## 4.3 QUANTITATIVE EVALUATIONS

As shown in Table. 1, our method leads all metrics on the MonoCap and ZJU-MoCap datasets. LAST outperforms 3DGS-based models (3DGS-Avatar and Gauhuman) in all settings thanks to our latent structure learning, which enables our model to utilize dependency associations that are often overlooked by traditional methods, enhancing its ability to capture intricate details and variations. This results in superior PSNR and SSIM scores. AnimateNeRF and InstantAvatar are state-of-the-art methods in NeRF-based 3D human reconstruction and can achieve reasonable scores in some settings (*e.g.,* InstantAvatar achieves 13.35 LPIPS* on MonoCap) for their meticulous implicit neural network. But they also fail on some datasets (*e.g.,* InstantAvatar fails on ZJU-Mocap with 68.41 LPIPS*) for complexity in human poses. In contrast, LAST achieves the best scores under all settings. Our model relies on latent factors to represent 3D humans, allowing for more targeted adjustments during optimization and enhancing the model's adaptability to complex poses and delicate clothing details, the reduction in LPIPS demonstrates our model's capability to minimize perceptual discrepancies, ensuring a more visually coherent output. Moreover, unlike NeuralBody, LAST maps latent space into explicit space (key Gaussian points), This mapping reduces the ambiguity often associated with implicit representations, enabling more accurate reconstructions. Additionally, compared to NeuralBody adjusts the implicit representation directly, the structured update process informed by the learned dependency matrix ensures that adjustments to one latent factor consider the influence on others, minimizing potential distortions and artifacts.

Table 1: Quantitative comparison of our method and other baseline methods on the ZJU-MoCap and MonoCap datasets. We use bold font to highlight the best result and underline the second-best result of each metric. Our method achieves the best PSNR, SSIM, and LPIPS on both datasets. LPIPS* = 1000 × LPIPS.

| | ZJU-Mocap (%) | | | MonoCap (%) | | |
|---|---|---|---|---|---|---|
| | PSNR↑ | SSIM↑ | LPIPS*↓ | PSNR↑ | SSIM↑ | LPIPS*↓ |
| NeuralBody | 29.03 | 0.964 | 42.47 | 32.36 | 0.986 | 16.7 |
| AnimateNeRF | 29.77 | 0.965 | 46.89 | 31.07 | 0.985 | 16.68 |
| InstantAvatar | 29.73 | 0.938 | 68.41 | 30.79 | 0.964 | 13.35 |
| HuGS | 30.21 | 0.962 | 36.26 | 31.12 | 0.982 | 23.42 |
| 3DGS-Avatar | 31.61 | 0.969 | 31.54 | 32.89 | 0.984 | 15.62 |
| GauHuman | 31.72 | 0.968 | 30.73 | 33.45 | 0.985 | 12.43 |
| **Ours** | 32.21 | 0.972 | 28.52 | 33.72 | 0.986 | 11.32 |

## 4.4 ABLATION STUDIES

We conducted ablation studies from 377 sequences of the ZJU-Mocap dataset to evaluate the impact of components in our model. The results are summarized in Table 2 and illustrated in Figure 4.

**Latent Structure Learning:** As shown in Figure 4, removing the latent structure learning module leads to significant degradation in model performance. Without the correct prior knowledge from the pre-training stage, the subsequent progressive update strategy and Gaussian density adapter optimization fail to utilize the correct latent factors and dependencies. This results in noticeable distortion and blur-

Table 2: Ablation Study

| Methods | PSNR | SSIM | LPIPS |
|---|---|---|---|
| Our | 32.45 | 0.9763 | 28.43 |
| W/O DenAd | 31.68 | 0.9715 | 35.06 |
| W/O ProOpt | 31.44 | 0.9716 | 33.92 |
| W/O LanStru | 29.78 | 0.9623 | 44.54 |

ring in the rendered output. As shown in Table 2, The PSNR decreases from 32.45 to 29.78, the SSIM drops from 0.9763 to 0.9623, and the LPIPS increases from 28.43 to 44.54. This indicates that the model without latent structure learning fails to capture the correct latent factors and dependencies, leading to noticeable distortion and blurring in the rendered output.

**Progressive Update Strategy:** Figure 4 illustrates the effect of omitting the progressive update strategy. Without this component, the model focuses solely on global optimization and cannot fine-tune local details. Consequently, areas requiring detailed refinements, such as the fingers and clothing

belts, exhibit substantial blurring and distortion. As shown in Table 2, The absence of the progressive update strategy (W/O ProOpt) leads to a decrease in PSNR from 32.45 to 31.44 and a minor drop in SSIM from 0.9763 to 0.9716, while LPIPS increases from 28.43 to 33.92. This demonstrates that without progressive updates, the model struggles with fine-tuning local details, resulting in blurring and distortion, especially in areas requiring detailed refinement.

**Gaussian Density Adapter:** Figure 4 also demonstrates the consequences of removing Gaussian Density Adapter. Due to the unreasonable Gaussian points distribution, the resulting output shows discrepancies in body contours, with notable distortions, such as on the left shoulder. As shown in Table 2, Omitting the Gaussian Density Adapter (W/O DenAd) results in a PSNR of 31.68, an SSIM of 0.9715, and an LPIPS of 35.06. While there is a slight decrease in PSNR and SSIM compared to the full model, the LPIPS value increases, indicating noticeable discrepancies in body contours and distortions in specific regions.

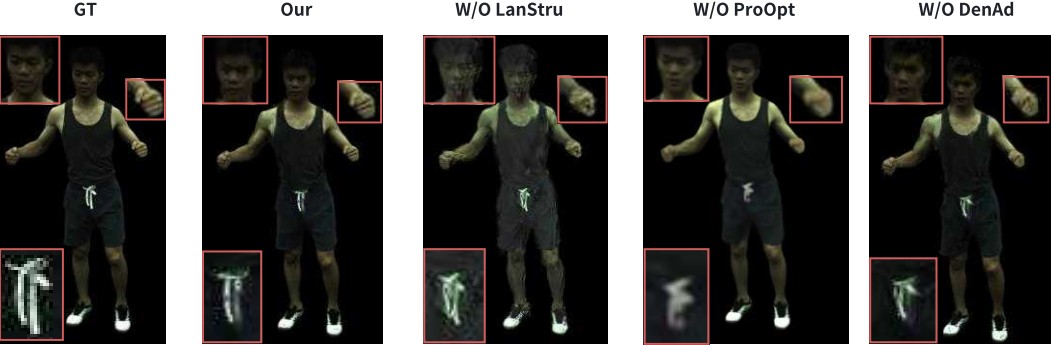

Figure 4: Ablation Study Results

## 5 CONCLUSION

We introduced an effective method for Gaussian human reconstruction, which disentangles latent factors with semantic information from visual features and addresses spurious correlations problem of dependencies between latent factors. Our method also proposes an innovative progressive update strategy. This strategy incorporates dependency ordering for hierarchical updates during optimization. This approach achieves a balance between local details and the overall result. Through extensive experiments, we validated the effectiveness of our proposed method. It is particularly effective in challenging scenarios involving complex poses and intricate clothing details. This innovative method provides insights into solving long-standing reconstruction challenges. For instance, in scenarios involving multiple people interacting or object occlusions, dependency inference can help the model better understand and handle dependency between objects. This improves the model's performance in reconstruction and reduces noise and inaccuracies during the process. In 4D dynamic scenes, where objects and human poses change over time, counterfactual inference can simulate and predict the effects of these dynamic changes. By considering "what-if" scenarios, the model can better understand and reconstruct object poses and interactions at different time points.

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

# A APPENDIX

## A.1 PRELIMINARY

**Human Reconstruction from Sparse Video**: Given a set of sparse video frames $V = \{v_1, v_2, \ldots, v_n\}$, where each frame $v_i$ contains a 2D image of the human body at different viewpoints, the objective is to generate an accurate $Y = \{y_1, y_2, \ldots, y_n\}$ that accurately represents the shape and pose of the human body, where $y_i$ rendered result at viewpoint $i$. Our goal is to minimize the difference between the rendered result of $y_i$ and the ground truth $v_i$.

**Latent structure learning**: In this work, latent structure learning is defined as the process of uncovering the $N$ latent factors that influence the effectiveness of 3D human reconstruction, denoted as $\mathbb{Z} = \{z_1, z_2, \ldots, z_n\}$, along with the dependencies $\epsilon$ between them, represented by $A$, from the observational data $X$. Each latent factor $z$ represents a specific characteristic of the human model (e.g., lighting, clothing, posture). We use the bold letter $z$ to denote the vectorial representation of $z$. Each element $A_{i,j} \in \{0, 1\}$ in $A$ represents whether the $i$-th latent factor $z_i$ has a direct effect on the $j$-th latent factor $z_j$.

**3D Gaussian Splatting**: 3D Gaussian Splatting (3DGS) is a 3D representation that approximates a continuous volume distribution by discretizing it into Gaussian points, which encapsulate attributes related to geometry (Gaussian scales) and appearance (opacities and colors). 3DGS begins with an initial sparse set of points derived from Structure from Motion (SfM) and refines this representation by controlling the density and attributes of the Gaussian points. Specifically, 3DGS constructs a loss function based on the difference between the rendered image $X_{render}$ and the ground truth image $X_{gt}$, optimizing the geometric features and color properties of the Gaussian points through backpropagation. Additionally, 3DGS determines whether to clone or split a point by evaluating the average loss gradient magnitude of the points in Normalized Device Coordinates (NDC), facilitating the transition from an initial sparse set of Gaussians to a denser configuration that more accurately represents the scene.

## A.2 HYPERPARAMETER ANALYSIS

### A.2.1 NUMBER OF LATENT FACTORS

The number of latent factors determines the model's ability to capture all relevant latent factors associated with 3D reconstruction and their dependencies. In this experiment, we tested the impact of different numbers of latent factors on the model performance using the ZJU-Mocap dataset. As shown in figure 5, the model performs best with 4 latent factors. When the number of latent factors is too low, the model may fail to capture all relevant latent factors in the 3D reconstruction process, and the dependency between these factors might be overlooked, leading to ineffective modeling of complex latent structures. Conversely, when the number of latent factors is too high, the model may learn many redundant or meaningless latent factors and dependencies, causing the dependency between factors to become vague and unclear.

### A.2.2 DENSITY CONTROL THRESHOLD

The density control threshold determines the minimum value for splitting or cloning Gaussian points during the 3DGS process, affecting the final density of Gaussian points and the reconstruction accuracy. In this experiment, we tested the impact of different density control thresholds on the model using 377 sequences from the ZJU-Mocap dataset.

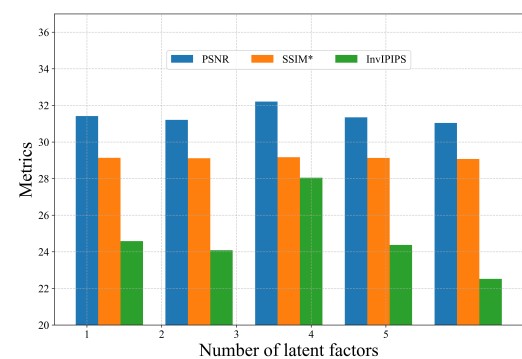

Figure 5: Effect of the Number of Latent Factors on Model Performance

We found that if the density control threshold, $\tau_{rec}$, is set too high, the splitting or cloning of Gaussian points may be suppressed, leading to insufficient Gaussian points in regions that require high detail. Conversely, if $\tau_{rec}$ is set too low, Gaussian points may split or clone excessively, causing the model to overfit local details.

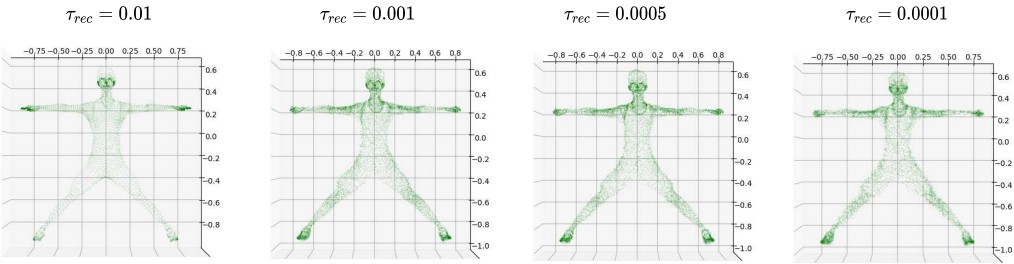

Figure 6: Effect of Density Control Threshold on Gaussian Point Density and Model Performance

As shown in the figure 6, when $\tau_{rec} \geq 0.01$, the splitting and cloning of Gaussian points are significantly suppressed, resulting in insufficient Gaussian point density in areas requiring rich detail (such as clothing and shoes). When $0.005 \geq \tau_{rec} \geq 0.001$, the model achieves a better balance, with splitting and cloning of Gaussian points controlled within a reasonable range. This allows the Gaussian point density to meet the requirements for detailed representation while avoiding excessive local overfitting. When $\tau_{rec} < 0.0001$, the splitting and cloning of Gaussian points occur too frequently, leading to potential redundancy in some regions (*e.g.,* noticeable blurring in facial areas).

### A.2.3 PROGRESSIVE UPDATE THRESHOLD

The progressive update threshold $\lambda$ determines the magnitude of adjustments made to the child latent factors during each update. The threshold essentially balances global optimization (parent latent factors) and local optimization (child latent factors). In this experiment, we tested the impact of different threshold $\lambda$ on the model's performance using the ZJU-Mocap dataset.

For samples with significant distortion in a single viewpoint, we selected different progressive update thresholds $\lambda$ for correction to verify the impact of different progressive update thresholds $\lambda$ on the overall performance of the 3D model. When the threshold is too large ($\lambda \geq 0.5$) led to higher reconstruction errors, suggesting that the updates were too aggressive, As the number of iterations increased, the impact of local adjustments on the overall model grew, disrupting the balance between global and local optimization. When the threshold is too small ($\lambda < 0.05$), the adjustments of child latent factors are insufficient to address the dependencies between parent latent factors and child latent factors, leading to poor convergence. When the threshold is moderate($\lambda = 0.05$), the adjustments to the child latent factors are neither too aggressive nor too minimal. This allows for

a balanced optimization where the local adjustments address the dependencies without disrupting the overall global optimization. When the threshold is optimal($\lambda = 0.1$), it ensures an optimal balance between the updates to the parent latent factors and the child latent factors. This optimal threshold allows the model to fine-tune the latent factors, leading to minimal reconstruction errors. The iterative process converges efficiently, resulting in high-quality 3D reconstructions.

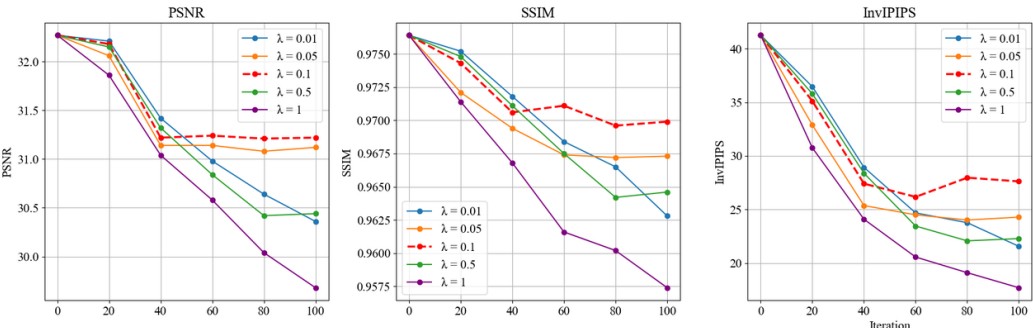

Figure 7: Effect of Progressive update threshold on Gaussian Point Density and Model Performance

