# OpenReview forum: "LAST: Latent Structure guided Gaussian Splatting from Monocular Human Videos"
_ICLR.cc/2025/Conference — ICLR 2025 Conference Withdrawn Submission_

### Official Review · Reviewer_u7Uj · 2024-10-24

**Soundness:** 3
**Presentation:** 3
**Contribution:** 2
**Rating:** 3
**Confidence:** 4

**Summary:**

This paper introduces the LAST framework for 3D human reconstruction from sparse views. It leverages a pre-trained Image-to-Point (I2P) model to augment the 3D Gaussian Splatting (3DGS) optimization pipeline. A specific latent structure is used to learn a dependency matrix, which captures the relationships between latent factors. By disentangling latent factors and modeling realistic dependencies from the input video frames, the framework enables dynamic adjustments to the density and attributes of Gaussian points during optimization. The framework achieves the following:

- Decouples latent factors from images (where each factor represents distinct semantic information), providing new supervisory signals for optimization. This leads to a clearer understanding of how each latent factor influences reconstruction.
- Uses the decoder to map the latent factor space directly into the point cloud space, creating a direct link between 3DGS and the latent factors.
- Establishes an updated sequential chain based on the topological order derived from the dependency matrix, minimizing interference among latent factors.

Several experiments are conducted to validate the proposed method.

**Strengths:**

The paper is well-written and presents a clear methodology, which is easy to follow.
Experimental results seem convincing, which validate the effectiveness of the method to some extent.

**Weaknesses:**

The main concern is the level of novelty. The efficiency of Gaussian Splatting is well established, and both 3DGS and 2DGS have been applied in human modeling. Additionally, the disentangling of factors such as lighting and pose is not a particularly novel contribution in this field. While incorporating semantic information into 3DGS can be beneficial by providing strong guidance, this approach depends heavily on the causal learning module (e.g., dependency graph). It remains unclear how robust and easy this pipeline is to train.

How resilient is the pipeline to unreliable causal inference results, what if the latent structure is unreliable (noisy)?

The paper would benefit from a discussion of its limitations and future work.

Furthermore, I could not find the supplementary video, making it difficult to evaluate the rendering quality.

## Minor Comments
The sentence: “Recent advances in 3D Gaussian Splatting, it is possible to achieve high expressivity with significantly faster training and rendering speeds compared to NeRF-based methods” is awkwardly phrased. Consider revising it for smoother readability.

**Questions:**

The claim: “However, existing methods lack research in 3D visual information” is questionable. While I am not deeply familiar with Causality in Vision, when it comes to 3D representations, approaches like NeRF combined with 3DGS and vision foundation models for scene graphs seem relevant. For broader context, consider recent works:

- End-to-End 3D Dense Captioning with Vote2Cap-DETR, CVPR 2023
- CAP3D: Scalable 3D Captioning with Pretrained Models, NeurIPS 2023
- ConceptGraphs: Open-Vocabulary 3D Scene Graphs for Perception and Planning, ICRA 2024
- LangSplat: 3D Language Gaussian Splatting, CVPR 2024
- LERF: Language Embedded Radiance Fields, ICCV 2023

---

### Official Review · Reviewer_fW61 · 2024-11-02

**Soundness:** 2
**Presentation:** 2
**Contribution:** 2
**Rating:** 3
**Confidence:** 4

**Summary:**

The proposed method aims to achieve realistic 3D human representation from sparse video data. The authors attempt to disentangle latent factors and remove spurious dependencies between these latent factors. The method also proposes a progressive update strategy that incorporates dependency ordering for hierarchical updates. Experimental results, in particular quantitative ones, validated the effectiveness of the proposed method. Ablation studies were conducted.

**Strengths:**

- (Originality) The proposed method seems fairly original due to its use of disentangling latent factors.

- (Significance) Authors evaluated the proposed method on various datasets (i.e. ZJU-MoCap, MoCap, and DNA-Rendering) and showed that their method achieved state-of-the-art performance.

- (Significance) In particular, the quantitative results (Table 1) are strong.

- (Significance) The benchmarks used for comparison are recent and up-to-date.

**Weaknesses:**

- Despite the explanations provided by the authors, it is hard to comprehend what the motivations for the proposed method are. For example, we are asked to visualise how wrinkles correlate with body occlusion. It will be more convincing if authors can provide a diagram instead.

- After repeatedly reading the motivations/explanation provided by the authors, I remain unconvinced on why there is a need to disentangle the different latent factors. Will be good to show concrete examples.

- Qualitative results are not very significant. In Fig 2, the proposed method indeed demonstrate improvement of results, but the improvement is not significant.

- Why is HuGS missing in Fig. 3 when it outperformed Gauhuman in Fig. 2. The quantitative results in Fig. 3 show improvement of the results, but again the improvement is not very obvious.

- Quantitative results for the DNA-Rendering dataset is not shown.

**Questions:**

Please address the weaknesses that I listed above. To summarize, I would like to see 1. concrete examples on why there is a need to disentangle the different latent factors. 2. Stronger qualitative results 3. Quantitative results for DNA-Rendering dataset

---

### Official Review · Reviewer_mMch · 2024-11-03

**Soundness:** 3
**Presentation:** 3
**Contribution:** 3
**Rating:** 6
**Confidence:** 4

**Summary:**

In this paper is presented a novel method for Gaussian human reconstruction from 2D images, that disentangles several latent factors with, apparently, semantic information, a key factor to achieve more accurate representations. To this end, a strategy to incorporate dependency ordering for hierarchical updates in optimization is proposed. Experimental results are provided in three datasets, including both quantitative and qualitative evaluation, and some comparisons with competing approaches.

**Strengths:**

The method in this paper is composed of two parts: a pre-trained image-to-plane model and an enhanced 3D Gaussian Splatting optimization module. In particular, the second module is informed by the prior knowledge acquired in the first one.  Particularly, a point cloud is generated from latent factors disentangled from observed data, by exploiting in training an encoder-decoder architecture. The authors incorporate intervention into the VAE training process.

Despite explaining this section well, including most of the details to understand the contribution, all the ingredients are well known in literature. In the second block, a Gaussian splatting approach is considered where the supervisory signals for density control are transformed from the pixels in images to the latent factors of the 3D model, that were obtained in the previous block. The point cloud density is adjusted based on their dependencies rather than in isolation.

In general, the paper is well written and is clear enough. The problem is challenging.

**Weaknesses:**

The paper proposes a method for reconstruction, but the authors exploit a ground truth of the 3D human representation. In my opinion this is a strong prior in this type of approaches, as the concept of reconstruction is not completely true.

Most comparisons are made using nerf-based methods. I think the authors should consider more 3GS approaches, discussing the main differences from a theoretical point of view and how those differences can be seen in the experimental results.

Comparison in Fig. 2 is partial. Just a 2D view is displayed. The authors should show the full 3D estimation they obtain by providing a couple of views where this information is clearly observed. As the data loss is to penalize the differences between the original image and the rendered one, of course, this estimation is good.

Some extra qualitative evaluations could help the reader: scenarios with multiple people, deformations, and so on.

The quantitative evaluation seems somewhat limited.

**Questions:**

Could the authors obtain an interpretation of the dependency associations that their method is learned? Why are the authors so sure that other methods are not able to learn them? Perhaps these associations are more difficult to interpret in other methods.

It is true that the quantitative results are better than those reported in the literature, but to be honest, the differences are not very large. So, can the authors maintain their claim that their method is able to extract latent information that the other methods do not?

A priori, the authors had more space to include more analysis and explanations, which unfortunately, they had not used.

---

### Official Review · Reviewer_p1cf · 2024-11-04

**Soundness:** 3
**Presentation:** 2
**Contribution:** 2
**Rating:** 3
**Confidence:** 4

**Summary:**

This paper proposes a method called LAST to reconstruct 3D humans from videos. At the core of the method are an I2P network to construct a point cloud from images, and a 3D Gaussian Splatting module to render humans. Experiments are conducted on three datasets including ZJU-MoCap, MoCap, and DNA-Rendering to verify the proposed method.

**Strengths:**

The paper proposes an I2P network to produce a point cloud from 2D images, which proves to enhance the optimization of 3D Gaussian Splatting. The idea of extracting a high-dimensional latent vector rather than just a point cloud is interesting compared with exisitng baselines that utilize SFM for Gaussian optimization. Expreiments and ablation studies are conducted on different datasets to verify each component of the proposed method.

**Weaknesses:**

1. 	The method part should be clarified. How to define the parent and child latent? Is this related to the kinematics chain? What are the explicit semantics that I2P produces? How the I2P network was trained?
2. The paper claims that the method can handle dynamic postures, but does not provide dynamic rendering results to verify the temporal consistency and generalizations on new poses. The rendering results are not of high quality, such as Fig. 3 and 4.
3. 	The rendering results on ZJU-MoCap are not at high-quality. The test poses/viewpoints of each method seem not the same, i.e., poses (Ours vs. GT) for the first human subject (e.g., 377), and camera viewpoints (Ours vs. Neural Body) for the second subject.
4. 	Some experimental details are missing. What are the training/test split, and rendering resolutions on the three datasets? The rendering results in Fig. 4 appear to be at a very low resolution.

**Questions:**

1. How about the training and inference time compared with existing approaches?
2.  Qualitative results: How about the temporal consistency? A video demo will be helpful.
3.  How about the generalizations on novel poses? Baseline method AnimateNeRF is animatable. Is LAST animatable?

---

### Note · Authors · 2024-11-13

I have read and agree with the venue's withdrawal policy on behalf of myself and my co-authors.